# Primitive Compartmentalization for the Sustainable Replication of Genetic Molecules

**DOI:** 10.3390/life11030191

**Published:** 2021-02-28

**Authors:** Ryo Mizuuchi, Norikazu Ichihashi

**Affiliations:** 1Komaba Institute for Science, The University of Tokyo, Meguro, Tokyo 153-8902, Japan; 2JST, PRESTO, Kawaguchi, Saitama 332-0012, Japan; 3Department of Life Science, Graduate School of Arts and Science, The University of Tokyo, Meguro, Tokyo 153-8902, Japan; 4Universal Biology Institute, The University of Tokyo, Meguro, Tokyo 153-8902, Japan

**Keywords:** origins of life, compartments, replication, evolution, RNA, parasites

## Abstract

Sustainable replication and evolution of genetic molecules such as RNA are likely requisites for the emergence of life; however, these processes are easily affected by the appearance of parasitic molecules that replicate by relying on the function of other molecules, while not contributing to their replication. A possible mechanism to repress parasite amplification is compartmentalization that segregates parasitic molecules and limits their access to functional genetic molecules. Although extent cells encapsulate genomes within lipid-based membranes, more primitive materials or simple geological processes could have provided compartmentalization on early Earth. In this review, we summarize the current understanding of the types and roles of primitive compartmentalization regarding sustainable replication of genetic molecules, especially from the perspective of the prevention of parasite replication. In addition, we also describe the ability of several environments to selectively accumulate longer genetic molecules, which could also have helped select functional genetic molecules rather than fast-replicating short parasitic molecules.

## 1. Introduction

Sustainable replication and evolution of genetic molecules are crucial steps for the emergence of life. A major obstacle in these steps is the appearance of parasitic molecules that replicate by exploiting the function of non-parasitic molecules, while not contributing to their replication [1,2]. To our best knowledge, the only effective means to circumvent the surge of parasites is compartmentalization that separates non-parasitic molecules from parasitic ones, and hence enables selective replication of non-parasitic molecules [1,2]. The question then is, what kind of compartments could have assisted the evolution of genetic molecules by avoiding parasite amplification on early Earth? Although lipid membranes serve as boundaries in all extant cells, more primitive materials or environments may have provided a similar effect. To date, Monnard and Walde presented a general overview of primitive types of compartmentalization, including non-lipid compartments that range from completely inorganic structures to biomolecular vesicles [3]. They described the sources, mechanisms of formation, and some functions of such compartments, but their potential roles in sustainable replication of genetic molecules remain to be reviewed. More recently, Wachowius et al. and Joyce and Szostak examined, albeit only briefly, the roles of some non-lipid compartmentalization on sustainable genetic replication [4,5]. Here, we specifically review theoretical and experimental advancements on how and what kind of primitive compartmentalization could have allowed the early evolution of life in the presence of parasitic molecules.

Contemporary cellular membranes mainly consist of phospholipids. However, due to the complicated process of phospholipid synthesis and the impermeability of phospholipid membranes to charged molecules in the absence of transporter systems, it is plausible that more ancient types of compartments preceded the advent of phospholipids in prebiotic evolution [5,6,7]. Single hydrocarbon chain amphiphiles (SCA), including fatty acids, have often been proposed as one of the first components of protocellular membranes, because of their permeability to small molecules and relatively high availability on early Earth by prebiotic chemical reactions or meteorites [5,6,7]. However, the SCA-based hypothesis is still problematic because of protocell instability, uncertainty of a robust prebiotic pathway of SCA synthesis, and the requirement of high SCA concentration for assembly [5,6]. Alternatively, more accessible non-lipid organic compounds or simple geophysical environments may have provided non-membranous compartmentalization. Some non-membranous compartments are advantageous over lipid membranes in that they can inherently concentrate long genetic molecules.

This review focuses on the possible primitive types of compartmentalization, especially non-lipid compartments, that could facilitate sustainable replication and evolution of genetic molecules. We also mainly focus on RNA, a likely candidate for a primitive genetic molecule, as it could act as a catalyst while storing genetic information [8,9,10]. In Section 2, we first summarize the roles of compartments for sustainable replication of genetic molecules by preventing parasite replication. Related to this, we also introduce the common “survival of the shortest” problem; parasites would evolve to replicate faster by losing unnecessary genetic regions for their replication and cause a problem even in the presence of compartments. In Section 3, we review ancient environment candidates that could allow the sustainable replication of genetic molecules. We describe nine types of conceivable compartmentalization: mineral surfaces, active fluid environments, crowded environments, ice eutectic phases, membranous compartments, membraneless cell-like compartments, gas bubbles, atmospheric compartments, and porous structures of hydrothermal vents. We particularly delve into minerals, ice, membraneless compartments, and hydrothermal vents, because a growing body of research has supported their validity for not only promoting the selection of functional genetic molecules, but also supplying RNA and possibly facilitating a relatively smooth transition to extant lipid-based compartmentalization. We review both experimental and theoretical studies. It should be noted that most theoretical research assumed an abstract model of molecular replicators that do not mimic detailed biochemical features of nucleic acids.

## 2. Parasitic Molecules and Prevention of Their Replication through Compartmentalization

Here, as one of the simplest primitive genetic molecules, we consider an RNA that has a dual role as a catalyst and a template. The RNA folds into a catalytic form (i.e., ribozyme) to replicate other unfolded RNAs (i.e., template) through ligation or polymerization of nucleotides (Figure 1a). During replication, mutations are introduced into RNA via ligation or polymerization errors. Most mutations would abolish the catalytic activity of a functional RNA, while maintaining its ability to act as a template, and therefore, the mutant RNA becomes parasitic because it does not catalyze RNA replication but its replication relies on the function of other RNAs (Figure 1b). Parasitic RNAs become a burden for the replication of functional RNAs, because functional RNAs would have to catalyze parasite replication together with their own replication. In addition, parasitic RNAs easily reduce their size by deletion, recombination, or inaccurate termination, because they do not need to maintain the information for catalytic activity anymore. Because short sequences typically replicate faster than longer ones, short parasitic RNAs would quickly outcompete functional RNAs, unless some kind of selection favors the catalytic activity (Figure 1b). This “survival of the shortest” phenomenon was first observed more than 50 years ago by Mills et al. [11] and has remained a major evolutionary problem in the field of the origins of life.

A means to prevent the replication of parasites is compartmentalization of functional RNAs from parasitic ones, by which functional RNAs can catalyze their own replication without parasite amplification (Figure 1c). The replication of parasites is repressed regardless of their lengths if they are segregated from functional RNAs. This evolutionary role of compartments has been proposed since the 1970s [1,2,12], and experimentally verified recently by us and other groups using artificial RNA replicator systems combined with Qβ replicase (and a cell-free translation system in some cases) [13,14,15]. These compartmentalized RNA replicators further underwent rich evolutionary phenomena, including adaptation [16,17], diversification [17,18], coevolution between functional (protein-coding) and parasitic (non-coding) RNAs [15,18,19], and coevolution between cooperative (different protein-coding) RNAs and the reinforcement of cooperation [20]. The compartment used in these experiments was a water-in-oil emulsion, a cell-like structure that was probably absent in ancient Earth. As described in Section 3, more rudimentary compartmentalization would have played a similar role and supported the replication of genetic molecules in the presence of parasites.

In addition to the above effect of compartments, in some cases, mechanisms for selecting longer RNAs would also be advantageous when short parasites appear. For example, once a parasitic RNA becomes short and excessively replicates with a functional RNA in the same compartment, it would be difficult to separate them in different compartments, except through strong dilution or degradation of RNAs (which we also discuss in Section 3). In this case, a mechanism that favorably traps longer RNAs in a compartment would help the replication of functional RNAs (Figure 1d). This would be particularly important for cooperative RNA replication because dilution of RNAs would separate not only a functional RNA from a parasite but also cooperative RNAs from each other. Previous research showed that even when compartments are introduced, a set of cooperative RNA replicators can survive only in limited concentration ranges with the appearance of fast-replicating short parasites [20]. Cooperation between distinct genetic molecules has been considered crucial for an early replication system to gain multiple functions by circumventing excessive mutation accumulation on one molecule [1,10,21]. As described in Section 3, many types of primitive compartmentalization have an innate potential to select longer genetic molecules.

## 3. Primitive Compartmentalization

Before the advent of lipid membranes, how were primitive genetic molecules separated from parasitic molecules? Some ancient environments likely generated compartment-like structures naturally or with primordial organic materials, whereas other environments enabled compartmentalization by limiting the diffusion of molecules to restrict the interaction of replicators in nearby molecules. Here, we classify possible primitive compartmentalization into three classes depending on the absence, presence, or partial presence of clear boundaries that separate individual compartments or groups of genetic molecules explicitly, as shown in Table 1.

In the first class, without clear boundaries, molecular diffusion is limited by several mechanisms; therefore, replicators form clusters that are spatially separated from each other. This class includes mineral surfaces, active fluid environments, and crowding environments. In the second class, with boundaries, groups of genetic molecules are separated distinctly. In this case, the boundaries need to be semi-permeable or undergo repetitive disruption and formation in order for genetic molecules to continuously access substrates for replication. The types (and respective boundaries) of compartmentalization in this class are ice eutectic phases (ice), membranous compartments (membranes), membraneless cell-like compartments (distinct aqueous phases), gas bubbles (aqueous phases), and atmospheric compartments (atmospheric gas phases). The third class is an intermediate of the first two classes, with incomplete boundaries, such as a solid boundary containing pores. Incomplete boundaries allow for inter-compartmental, although limited, diffusion of genetic molecules. This class includes the porous structures of hydrothermal vents with mineral-based boundaries, in which porous structures separate individual compartments (aqueous phases) in the vents. Such structures are distinguished from another mineral-based environment, i.e., mineral surfaces in the first class, because the group of genetic molecules could be separated without the limitation of molecular diffusion due to their interaction with mineral surfaces.

Another important aspect of compartmentalization is how new clusters of replicators or compartments could be generated. For sustainable replication of genetic molecules, functional molecules keep accessing parasite-free locales because parasites are continuously produced by mutations. This can be achieved by the continuous regeneration of new compartments. Compartments may be constantly synthesized in some environments, as briefly described in each subsection. However, even if the number of compartments (or spatial locals) is unchanged, parasite-free locales could also be generated by the degradation or dilution of parasites.

### 3.1. Compartments without Boundaries

Separation of parasitic molecules from non-parasitic molecules can be achieved without clear boundaries, if diffusion of genetic molecules is sufficiently slow (Figure 2a). Several factors could limit diffusion of genetic molecules, including their attachment to solid surfaces (e.g., mineral surfaces), active flows in aqueous phases, and highly crowded environments.

#### 3.1.1. Mineral Surfaces

Mineral surfaces were ubiquitous throughout early Earth and likely played diverse roles in the origins of life [60]. More than 30 types of minerals, including clay, sulfide, oxide, carbonate, and oxide minerals, have been shown to accumulate RNA molecules on their surfaces (Figure 2b, left) [22,61,62]. Furthermore, in 1980, Gibbs et al. showed that hydroxyapatite minerals selectively adsorbed longer oligoadenylates, from 2–22 nt sequences [23]. We later showed that this phenomenon is general by demonstrating that a wide variety of minerals with different chemical compositions (pyrite, pyrrhotite, magnetite, calcite, and hydroxyapatite) favorably bind longer RNA molecules better than shorter ones from a pool of 8–24 nt random sequences [22]. A mathematical model further showed that enrichment of longer RNAs on mineral surfaces is thermodynamically favorable because a longer RNA has a larger adsorption energy [22]. Such an inherent property of minerals may help propagate RNA replicators in the presence of short parasites.

Although no research has successfully demonstrated RNA replication on mineral surfaces, the potential of mineral surfaces (or equivalent two-dimensional surface structures that accumulate genetic molecules) to select functional genetic replicators has long been studied theoretically, especially by the groups of Hogeweg, Szathmary, and Higgs [24,25,26,27,28,29,30,31,32,33]. These studies generally assumed a two-dimensional grid with each cell containing only one or a few replicators (Figure 2b, right). In these models, the interaction and diffusion of replicators occurred only in nearby sites, because the mobility of RNA bound onto mineral surfaces is expectedly slow [63]. Such a limited dispersal causes sustainable clustering of related replicators (as if they were compartmentalized), leading to selective replication of non-parasitic replicators [24,25,26,27,28,29,30,31,32,33]. A set of distinct cooperative replicators can also be selected [24,25,27,28,32]. Some of the research further demonstrated the evolution of replicator-associated parameters in surface models [26,28,29,31].

Importantly, various minerals could also have helped supply RNA on early Earth by several mechanisms. Minerals helped assemble activated nucleotides either with or without template RNA [64,65,66,67] and protect RNA from degradation [62,68]. Minerals also supported the synthesis of RNA components [69,70,71,72]. It has been further shown that several ribozymes maintained their activities in the presence of mineral surfaces [22,68,73,74]. In addition, diverse types of mineral surfaces assisted the formation of primordial lipid (fatty acid) vesicles, some of which encapsulated RNA attached with mineral particles [34,35,36]. Thus, it is conceivable that RNA replicators flourished on mineral surfaces first and were later removed from these surfaces, perhaps in combination with the emergence of lipid vesicles.

#### 3.1.2. Active Fluid Environments

Molecular diffusion can also be limited by processes other than surface adsorption. Krieger et al. theoretically demonstrated that active fluids in aqueous environments (such as based on winds near the surface of the ocean or ponds) could regulate the diffusion of molecular replicators by trapping them in small spatial domains, known as Lagrangian coherent structures (LCS) (Figure 2c) [37]. They demonstrated that apparently isolated LCS slowed down the diffusion of trapped replicators and further behaved like distinct compartments as they underwent fusion and division, although some migrations of replicators between LCS were also observed. Combining these mechanisms, the study showed that LCS helped functional replicators to flourish in the presence of parasites. An experimental model of LCS with RNA or other genetic molecules remains to be designed in the future.

#### 3.1.3. Crowded Environments

In principle, the extent of molecular crowding also influences the diffusion of genetic molecules [75]. Kamimura and Kaneko’s theoretical study considered two types of replicators that catalyze the replication of each other and demonstrated the selection of functional replicators in crowding environments (Figure 2d) [38]. If mutual replicators have greatly different replication rates and stabilities, faster and less stable replicators (X) would surround a slower and more stable replicator (Y), thereby forming multiple molecular clusters. These clusters further behaved like compartments as they underwent a fission-like phenomenon in synchronization with the replication of a slow replicator and its Brownian motion; molecular clusters composed of only non-parasitic molecules can continuously grow. A parasite of Y (with similar replication and degradation rate) could also be selectively diffused out of the cluster as it does not contribute to the replication of X. Further studies would be expected to test whether sufficient crowdedness for these phenomena could be provided in prebiotic environments.

### 3.2. Compartments with Boundaries

This class of compartments has boundaries that can segregate non-parasitic molecules from parasites explicitly, thereby enabling more facile selection of functional genetic molecules. Various materials, such as ice, lipid membranes, and aqueous phases, can be used as boundaries in primitive environments (Table 1). In contrast to the open environments described in Section 3.1., the presence of boundaries may prevent genetic molecules from accessing small substrates that are required for continuous replication. Therefore, some mechanisms must support the inflow of small molecules in these compartments.

In addition, because boundaries prevent the outflow of genetic molecules, compartments should grow and divide or undergo equivalent phenomena to avoid the accumulation of too many parasites in a single compartment. In primitive compartments, these processes could have occurred by at least three mechanisms: cycles of disruption and formation (Figure 3a), fusion of multiple compartments and subsequent fission (Figure 3b), and growth and division of compartments (Figure 3c). The first two processes were often referred to as transient compartmentalization. Through these processes, genetic molecules in each compartment can be randomly redistributed into new compartments, by which functional genetic molecules could be separated from parasitic molecules if the number of parasites is not large compared with the number of newly formed compartments. Previous studies that used water-in-oil emulsions or theoretical models demonstrated the sustained replication of genetic molecules with the appearance of parasites through all three processes: cycles of disruption and formation (Figure 3a) [14,40,41,42], fusion and fission (Figure 3b) [13,15,20,43,44], and growth and division (Figure 3c) [29,45,46,47] of compartments.

#### 3.2.1. Ice Eutectic Phases

Ice structures were likely prevailing on early Earth [76,77]. When aqueous solutions comprising ions or other solutes freeze, ice crystals grow and exclude solutes in an interstitial brine, serving as a compartment-like structure known as the eutectic phase (Figure 3d, left). Attwater et al. showed that the eutectic phase can be separated into small disconnected microstructures scattered in ice crystals, depending on the type and concentration of solutes (e.g., magnesium chloride) [39]. They also showed that diffusion of a ~200 nt RNA polymerase ribozyme can be reduced by several orders of magnitude in the eutectic phase, which would also contribute to the compartmentalization of primitive genetic molecules. Attwater et al. further demonstrated in silico that fragmented structures such as those seen in ice crystals supported the replication of self-replicating species by neutralizing the effect of parasites that show faster replication and diffusion (Figure 3d, right) [39].

Closed compartments in ice crystals would probably prevent inter-compartment diffusion of solutes and genetic molecules, and hence, continuous genetic replication. However, cycles of disruption and reformation of compartments (Figure 3a), which can be caused by freezing and thawing (F-T) through day–night temperature cycling, could induce uptake of substrates. Such transient compartmentalization would also help to prevent accumulation of parasitic molecules in the compartments.

In addition, freezing or F-T cycles supported other important processes for the emergence of life, such as synthesis of RNA through non-enzymatic polymerization of activated nucleotides [78,79,80], formation of longer RNAs through non-enzymatic ligation and recombination of short fragments [81,82], and assembly and catalysis of ribozymes [39,83,84,85]. F-T cycles have also been demonstrated to induce the encapsulation of genetic molecules in phospholipid vesicles [48,49], their inter-vesicle exchange [86], and sustainable RNA replication through fusion-division of vesicles [87]. Thus, ice environments seem to be plausible sites for the development of early life.

#### 3.2.2. Membranous Compartments

Both lipid and non-lipid membranes can also serve as boundaries that separate functional genetic molecules from parasitic ones. In contrast to extant cellular membranes with protein transporters, primitive membranes should have been semi-permeable, so that encapsulated genetic molecules could access small substrates in the environment. As reviewed extensively elsewhere [5,6,7], semi-permeable membranes based on fatty acids or other simple hydrocarbon-based amphiphiles seem to be relatively promising candidates, because they are compatible with various RNA-related reactions and could undergo growth and division. Semi-permeable membranes can also form from diverse non-lipid building blocks [88], including inorganic nanoparticles and proteins. One of the prebiotically relevant examples is based on natural clay minerals, created by the assembly of clay nanoparticles on air-bubble surfaces and subsequent dissolution of the air phase [89], which may add another selective advantage on genetic replicators to mineral surfaces that by themselves could help evolve functional molecules, as described in Section 3.1.1.

#### 3.2.3. Membraneless Cell-Like Compartments

Since the proposal by Oparin [90], droplets formed by liquid–liquid phase separation (LLPS) (Figure 3e, left) have been considered as another candidate compartments before the origins of life. These membraneless cell-like compartments enable the sequestration of genetic molecules, while allowing free exchange of small molecules with the environment. Genetic molecules in LLPS droplets can therefore easily access substrates in the environment. Two types of LLPS droplets, coacervates (typically generated from the association of oppositely charged polyelectrolytes) and aqueous two-phase systems (ATPS, generated by the segregation of multiple polymers), have been particularly studied in the context of the origins of life [91,92,93,94], although coacervates may be more relevant to the origins of life, because they can be made of prebiotically more accessible materials such as mononucleotides and short peptides [95,96].

From the perspective of eliminating short parasitic molecules, Drobot et al. showed that coacervates based on carboxymethyl dextran sodium salt and poly-L-lysine can selectively accumulate 39-mer RNA molecules (a minimal hammerhead ribozyme) by preventing their diffusion into a surrounding environment, while permitting the exchange of 6- or 12-mer RNAs (substrates before or after cleavage by the ribozyme) with the environment (Figure 3e, center) [50]. The selective accumulation of longer RNAs was probably the result of their strong interactions with components of coacervates, and thus the selectivity varied depending on the sequence of RNA and the composition of coacervates [50]. In another study, Strulson et al. used an ATPS of polyethylene glycol (PEG) and dextran (DEX) to demonstrate the length-dependent partitioning of a set of ≤ 40 nt RNA molecules (generated by hydrolysis of a hammerhead ribozyme); the fraction of RNA localized in droplets increased exponentially as the length of RNA molecules increased [51]. Thus, both coacervates and ATPS have a certain advantage in selecting longer genetic molecules.

LLPS droplets seem to undergo all processes shown in Figure 3a, 3b, and 3c. Both coacervates and ATPS were shown to undergo a cycle of formation and dissolution (Figure 3a) by fluctuating environmental factors such as temperature, pH, and dryness [97,98,99]. In addition, both types of droplets spontaneously coalesce with other droplets and could undergo fission (Figure 3b), such as induced by a shear force (e.g., provided from the sea current). Droplets may also form, grow, divide, or dissolute (Figure 3a,c) in association with the generation or consumption of droplet materials [100,101]. These processes would help functional genetic molecules to segregate from parasitic molecules.

Furthermore, our group recently showed that a PEG/DEX ATPS can support the self-replication of a relatively long (2041 nt) artificial single-stranded RNA (with its encoded Qβ replicase subunit) by preventing the appearance of short (>~220 nt) parasitic molecules that lost a part of the replicase gene (Figure 3e, right) [52]. The suppressed replication of the parasites was due to the prevention of inter-droplet exchange of encapsulated RNAs, while another study showed that much shorter (15–50 nt) RNAs would be rapidly exchanged in a different composition of PEG/DEX ATPS [102]. The permeability of the ATPS to small molecules also enabled continuous RNA replication with the addition of nucleotides [52], which is another advantage of LLPS droplets.

LLPS droplets are also compatible with prebiotic RNA-associated reactions and formation of lipid vesicles. Multiple ribozymatic reactions and non-enzymatic RNA polymerization were demonstrated in either or both coacervates and ATPS with some enhancement [50,51,103,104]. Other studies also showed that fatty acid and phospholipid membranes self-assembled at the surface of coacervates and ATPS droplets, respectively [53,54]. These LLPS droplets may have facilitated the transition from primitive compartmentalization to lipid-based compartmentalization by acting as scaffolds for membrane formation.

Several other membrane-free compartments have also been proposed as primitive compartments, because they can be potentially generated from prebiotic materials. These examples include hydrogels [105,106], especially clay-based hydrogels [106], polyester-based droplets [107], and a new class of coacervates based on liquid crystals [108]. Further investigation regarding the usage of these compartments to select or evolve genetic molecules is required.

#### 3.2.4. Gas Bubbles

For the separation from parasitic molecules, genetic molecules should not necessarily be distributed inside compartments. For example, air–water interfaces can accumulate DNA [55,109,110] and RNA [55], and therefore, it is conceivable that gas bubbles dispersed in a water phase could act as distinct compartments that separate groups of genetic molecules from each other (Figure 3f). Morasch et al. demonstrated that gas bubbles in a hydrothermal environment favored the accumulation of longer (132 nt) single-stranded DNA molecules compared to shorter (15 nt) ones [55], thereby potentially contributing to the elimination of short parasitic molecules. The favorable accumulation of longer DNA molecules was attributed to their lower diffusion coefficients. They also showed that gas bubbles enhanced ribozyme catalysis and facilitated the encapsulation of DNA and RNA in lipid vesicles or vesicle aggregates by co-accumulating genetic molecules and lipid vesicles at the air–water interface. Genetic molecules at air–water interfaces would freely access substrates in the water phase. Bubbles probably do not last perpetually and would repeat cycles of disruption and formation (Figure 3a). The next step in this research direction is to demonstrate RNA replication at air–water interfaces.

#### 3.2.5. Atmospheric Compartments

Compartments could also exist in the atmosphere as water droplets or aerosols. In 1979, Woose proposed the possibility that water droplets in the air acted as cell-like compartments on early Earth, especially when most of the water was vaporized due to the high temperature of early Earth [111]. After formation of the ocean, lakes, or ponds, cell-sized aerosols would also have formed near water surfaces by wind actions and provided compartments (Figure 3g) [56,112]. An advantage of aerosols is the plausibility of the seamless transition to membrane-bound compartments, if lipid molecules are provided [56]. Aerosols can grow and divide into daughter particles through multiple mechanisms, while compartments would also not last perpetually and would eventually be destroyed (such as through deposition to the ocean) [112]. Although it should be experimentally tested in the future, these processes of reorganizing internal contents could help to select for non-parasitic genetic molecules.

### 3.3. Compartments with Incomplete Boundaries

This class of compartments has incomplete boundaries, such as solid phases containing pores observed at hydrothermal vents. An incomplete boundary allows genetic molecules to favorably interact with others in the same compartment, while limiting their migration to other compartments. The incompleteness of the boundary may facilitate the uptake of small substrates, which is an advantage over most compartments with complete boundaries, as described in Section 3.2., although selection for functional genetic molecules would be relatively difficult, because parasitic replicators could intrude the surrounding compartments.

#### Porous Structures of Hydrothermal Vents

Hydrothermal vents would have prevailed in the deep sea of early Earth and provided locales to initiate life-like processes [113,114]. Modern hydrothermal vents contain an interconnected network of inorganic porous structures (Figure 4a) [115,116], and it has been proposed that such naturally forming compartments could have functioned as protocellular structures in ancient vents [117,118,119].

The role of interconnected compartments in the selection of non-parasitic replicators has been investigated theoretically [47,58]. These models assumed a two-dimensional grid, similar to the surface model discussed in Section 3.1.1., but each cell could contain many replicators (Figure 4b). Each replicator can interact with others only in the same compartment but can also migrate to neighboring cells. Studies have demonstrated that restricted molecular diffusion allows the selection of functional replicators in the presence of parasites [47,58]. Selection of a set of cooperative replicators with different replication kinetics was also observed [58]. The number of successfully coexisting cooperators in the interlinked compartments seemed higher than in the surface model, where interaction between replicators was too limited [58]. However, compared with a model with boundaries described in Section 3.2., a lower mutation rate (lower frequency of parasite generation) seemed to be required for sustainable genetic replication in compartments with incomplete boundaries [47].

Moreover, the positive role of the compartment regarding the enrichment of longer DNA and RNA in hydrothermal environments (Figure 4c) has been examined empirically, especially by the Braun group. Thermal gradients generated by the temperature difference between a hot vent and cold water induce thermophoresis and convection through a compartment, both of which could drive genetic molecules in different directions and in combination concentrate them at a cold locale within the compartment, with higher efficiency for longer sequences [120,121]. Theoretically, the efficiency of accumulation scales exponentially with Soret coefficients (defined as thermodiffusion coefficients divided by diffusion coefficients) of genetic molecules [120,121]. Thermal convection in the compartment also provides genetic molecules with temperature cycling necessary for efficient replication by an RNA polymerase ribozyme [122]. By applying these principles to a hydrothermal open pore system, Kreysing et al. demonstrated the selective trapping of longer DNAs among 20–200 bp strands as well as the selective DNA amplification of 75 bp strands over 36 bp strands in the pore [57], possibly contributing to the elimination of short parasitic sequences. It should be noted, however, that these experiments were performed using synthetic devices specifically designed for these experiments. Recent progress in generating inorganic precipitate membranes that resemble those at hydrothermal vents [123] would allow us to explore whether the above findings are applicable to natural geochemical structures.

In addition, simulated hydrothermal environments generated short RNA oligomers from both activated and non-activated nucleotides [124,125], and facilitated the accumulation of fatty acids followed by vesicle formation and the encapsulation of genetic molecules [59]. These studies imply that hydrothermal vents are compatible with the supplementation of genetic molecules as well as the formation of organic compartments.

## 4. Conclusions and Perspectives

In this review, we introduced possible primitive compartmentalization that could allow the replication of RNA by segregating parasitic molecules, selecting longer sequences, and yet supplying substrates. As summarized in Table 1, at least nine types of such compartmentalization could have been available based on simple geophysical environments or primitive biological materials. To date, the effect of compartments on the evolution of genetic molecules has been mainly studied theoretically, but several recent experiments have also demonstrated the selection of longer RNA molecules and even selective RNA replication by preventing parasite amplification. The next key step is to demonstrate sustained RNA replication in various primitive compartments.

The different types of compartmentalization discussed in this review could have operated synergistically. For example, compartments of hydrothermal vents have mineral-based boundaries and likely contain gas bubbles [55]. Mineral-bound compartments can also form in other ways [89,126]. Upon the formation of compartments with fatty acid membranes, mineral particles can be encapsulated [34]. To date, neither theoretical nor experimental studies have investigated the influence of such combinatorial environments on the evolution of genetic molecules. Future studies should address these possibilities to better understand the roles of primitive compartmentalization in the early evolution of life.

Finally, the plausibility of each of the discussed environments on early Earth should be evaluated in light of geochemical and astrochemical knowledge [3]. In fact, most theoretical or experimental studies on the evolution of genetic molecules do not incorporate all the details on natural environments, in part because it is impossible to do so with the current computational or experimental resources. That said, the development of theoretical and experimental models that resemble an ancient environment, at least partly, provides insights into the early evolution of life and help refine ideas regarding likely sites for the emergence of life. Future collaborations between different research communities should lead to more plausible hypotheses for the origins and early evolution of life.

## Figures and Tables

**Figure 1 life-11-00191-f001:**
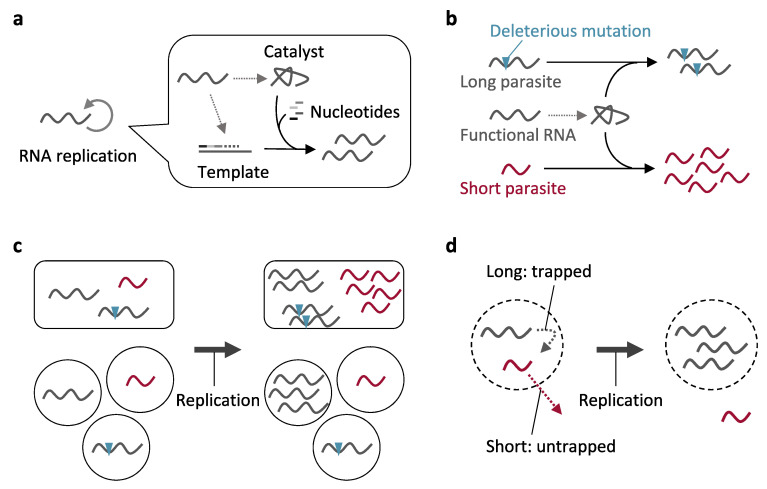
Prevention of parasite replication through compartmentalization. (**a**) A possible scheme of RNA replication. A functional RNA could either be a catalyst or a template through its folding or unfolding, respectively. A catalyst replicates a template through polymerization of nucleotides. (**b**) Replication of parasitic RNAs. A parasitic RNA with a deleterious mutation could replicate by exploiting the function of other RNAs, while not contributing to their replication. Parasite replication would be severely faster if a parasite becomes short by losing unnecessary genetic regions. (**c**) If functional and parasitic RNAs are separated into distinct compartments, only functional RNAs can replicate. (**d**) Compartments that selectively trap longer genetic molecules have an advantage for replication of functional molecules.

**Figure 2 life-11-00191-f002:**
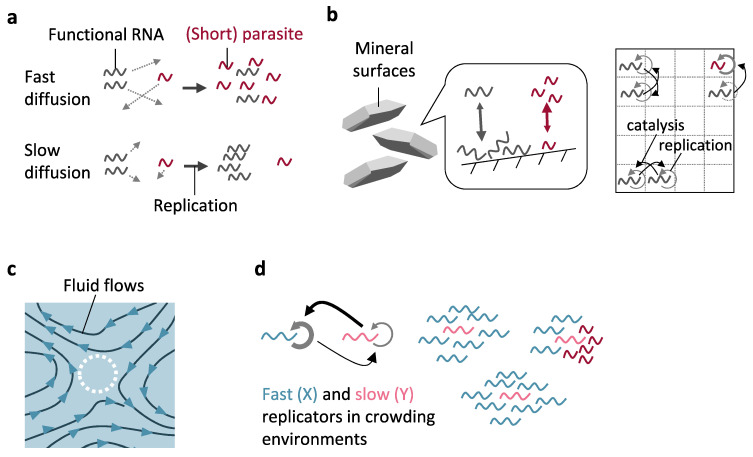
Primitive compartmentalization without boundaries. (**a**) If molecular diffusion is sufficiently fast, a functional RNA catalyzes replication of any RNA molecules, including parasitic ones. With limited diffusion, however, a functional RNA replicates only nearby molecules, such as its progeny. (**b**) Limited diffusion of genetic molecules can be achieved on mineral surfaces, which can also inherently select for longer RNA molecules (left). Theoretical studies based on two-dimensional surface models showed that the spatial clustering of replicators caused by limited diffusion sustained the replication of clusters consisting of functional replicators (right). (**c**) Active fluids in aqueous environments could regulate molecular diffusion especially by forming an apparently isolated region (dotted circle), supporting replication of functional replicators in the presence of parasites. (**d**) In highly crowded environments, a pair of mutualistic replicators with different replication rates and stability can be spatially separated from other sets, which prevents the spread of parasites.

**Figure 3 life-11-00191-f003:**
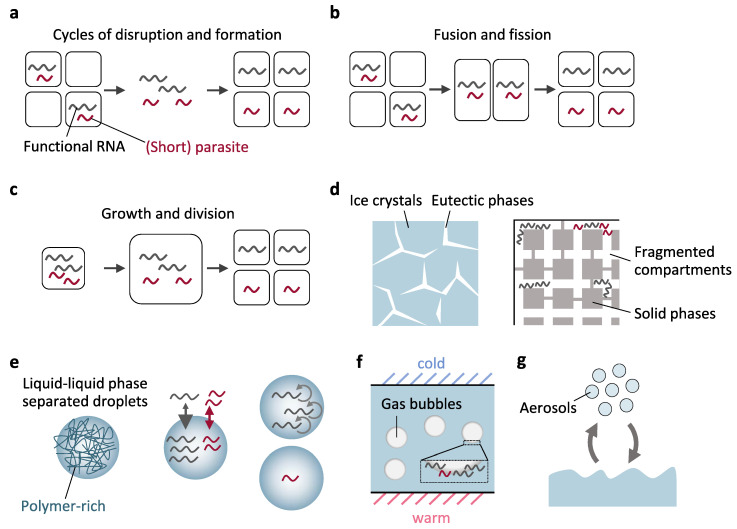
Primitive compartmentalization with boundaries. (**a–c**) Separation of functional genetic molecules from parasites through compartmentalization could be facilitated by multiple mechanisms: cycles of disruption and formation (**a**), fusion and fission (**b**), and growth and division of compartments (**c**). (**d**) Separated microstructures of eutectic phases could form in ice crystals (left). Molecular diffusion is also limited within each compartment. A theoretical model based on such fragmented compartments showed the selective replication of non-parasitic molecules (right). (**e**) Liquid–liquid phase separation (LLPS) droplets (left) favorably accumulated longer genetic molecules (center) and prevented replication of short parasitic molecules through compartmentalization (right). (**f**) Longer genetic molecules could accumulate in gas bubbles more efficiently than shorter ones at air–water interfaces, and gas bubbles may function as separated compartments. (**g**) Atmospheric compartments such as aerosols could also act as cell-like compartments.

**Figure 4 life-11-00191-f004:**
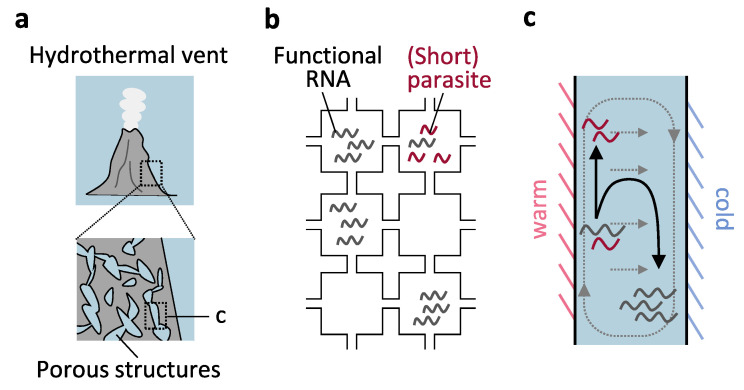
Primitive compartmentalization with incomplete boundaries. (**a**) Interconnected porous structures were found at hydrothermal vents. (**b**) Theoretical models indicated that restricted interactions between replicators and limited diffusion in such porous structures could allow for the selection of functional RNAs. (**c**) Thermophoresis and convection (grey dotted lines) induced by thermal gradients around a pore could selectively trap longer genetic molecules within a pore for amplification. An example of corresponding sites is indicated in panel a.

**Table 1 life-11-00191-t001:** Possible primitive compartmentalization.

Class (and Major Mechanism for Separating Non-Parasitic Molecules from Parasites)	Elements for Compartmentalization	Boundary	Additional Mechanism that Contributes to Segregating Parasitic Molecules	Mechanism of Substrate Uptake from Environments	Selective Accumulation of Longer Genetic Molecules	Selective Replication of Non-Parasitic Genetic Molecules		Possible Transition to Lipid Vesicles
						Using an Analogous Computational Model	In a Laboratory Experiment	
1. Without boundaries (limited diffusion)	Mineral surfaces			Diffusion	Yes [22,23]	Yes [24,25,26,27,28,29,30,31,32,33]		Mineral-assisted [34,35,36]
	Active fluid environments		Apparent group formation	Diffusion		Yes [37]		
	Crowded environments		Apparent group formation	Diffusion		Yes [38]		
2. With boundaries (group formation)	Ice eutectic phases	Ice	Limited diffusion	Disruption and reformation		Yes [20,29,39,40,41,42,43,44,45,46,47], but mostly by using general protocell models that have one or more features of each compartmentalizaiton with boundaries		Freeze-thaw-assisted [48,49]
	Membranous compartments ^1^	Membranes		Diffusion through semi-permeable membranes				
	Membraneless cell-like compartments ^1^	Distinct aqueous phases		Diffusion through semi-permeable interfaces	Yes [50,51]		Yes [52]	Interface-assisted [53,54]
	Gas bubbles	Aqueous phases		Diffusion	Likely ^2^ [55]			Interface-assisted [55]
	Atmospheric compartments ^1^	Atmospheric gas phases		Disruption and reformation				Interface-assisted [56]
3. With incomplete boundaries (limited group formation)	Porous structures of hydrothermal vents	Minerals		Diffusion through incomplete boundaries	Yes [57]	Yes [47,58]		Thermal gradient-assisted [59]

^1^ These compartments may reproduce. ^2^ Although the accumulation of the mixture of different lengths was not investigated, more effective accumulation was observed with longer sequences.

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
