# Peer review of "Primitive Compartmentalization for the Sustainable Replication of Genetic Molecules"

_life, 2021, doi:10.3390/life11030191_

Round 1
Reviewer 1 Report
Mizuuchi and Ichihasi present a very well-written review that comprehensively catalogs prebiotic compartment types that could have facilitated the retention of longer genetic polymers over their smaller counterparts. Such a feature would have enabled catalytic polymers to retain their evolutionary advantage, rather than having their fitness be compromised by parasitic shorter polymers.
The manuscript is a focussed and informative read. I recommend publication after the following minor changes:
- Describe in brief in the main text the mechanism for longer nucleic acid retention, where possible, whether it be slower diffusion times, stronger intermolecular forces with a surface or matrix (e.g. hydrogel), stronger surface-activity, or lower permeability.
- Change the repeated use of ‘boundary phase’. ‘Phase’ suggests something very specific - liquid, solid, gas - and is ambiguous in the way it’s currently used in the manuscript. ‘Boundary phase’ is used to describe a very important concept and delineate between different compartment types. Is a mineral surface not a boundary phase? Instead from Table 1 it is obvious that the distinction between the three categories is the mechanism for separating longer polymers from shorter ones (which I would word as such, rather than ‘non-parasitic vs parasites’). Consider rewording the 3 'classes of compartmentalization'.
Author Response
Please see the attached pdf file.

Reviewer 2 Report
The present review by N. Ichihashi and R. Mizuuchi provides an overview of possible environments for primitive compartmentalization of genetic material in order to achieve sustainable replication. The necessity to shield genetic material from parasitism utilizing compartmentalization takes a central role and a wide range of possible candidates of evolutionary early compartments are discussed. The different possible compartments are debated in regards of their composition, diffusion mechanism, boundary phases and capability to compartmentalize and accumulate long RNA polymers.
The review is adroitly written and gives a compelling overview of a broad range of possible compartments for genetic material. The review can be published after addressing the following issues:
- In Table 1. the distinction is made between classes of compartments without boundary phases, such as mineral surfaces, and compartments with incomplete boundary phases, such as “porous structures of hydrothermal vents made of minerals”. More emphasis should be put on the distinction between mineral surfaces and porous structures of hydrothermal vents made of minerals: how are they different? Why does one mineral surface have a (incomplete) boundary phase while the other mineral surface doesn’t?
- In Figure 1 a,b a solid arrow hints replication, while in figure 1 c,d replication is depicted by the word “replication”. Why not use solid arrows in c,d as well?
- In Figure 2b the right depiction (2d grid) is not very informative with the current description. The description talks about “selective replication of non-parasitic replicators…”, while no replication is actually depicted in the picture.
- In Figure 3d the right depiction (theoretical model) is not self-explanatory. What exactly is depicted with the connected squares? Why are the replicators next to those squares but not encapsulated (like in an eutectic phase?). What exactly is depicted by the connections between the squares?
- Transient compartmentalization is only mentioned in the context of F-T-cycles (line 271), but should be more explicitly discussed in the context of prevention of parasitism and relevance for the role on early earth as well as its general (possible) connection to metabolism.
- In this manuscript, the role of different compartments has been reviewed mainly focusing on the selective separation of non-parasitic molecules from parasitic ones. However, this selective separation can’t occur for some of these compartments (Table 1). Hence, it might be useful to clarify the survival pathways of functional RNA in these compartments and also discuss the potential role of compartments (in general) on host–parasite coevolution and diversification.
- Authors argue that compartments can prevent “the survival of the shortest”(line 96), but the exact mechanism by which this is achieve should be explained in more detail.
- In Fig.3 a-c, authors should explain the mechanism by which short RNA and long RNA are sorted (or “redistributed”) in completely separate compartments after disruption-formation cycles, fusion-fission and Growth-division, respectively.
- Even if the compartments (with boundary phase) can separate parasitic RNAs from longer RNAs, if they cannot reproduce themselves, they would not be able to survive and evolve. As a general point, it would be nice if the authors would discuss self-reproduction of compartments and their contents and the challenges this poses in more detail.
- I also suggest to add one table to show potential pathways for transition from each primitive compartment to membranous (modern) cells which is already discussed in different parts. The self-reproduction possibility of each compartment can be added to this table as well.
Some minor points of language:
- On line 57 “advantages” needs to be changed to “advantageous”
- On line 316 “the” is wrongly placed (unnecessary)
- On line 243 and 244, please refer first, second and third cases to appropriate schemes (Fig.3) or processes.
- On line 566 “self-replicatingmolecules” should be changed to “self-replicating molecules”
Author Response
Please see the attached pdf file.

Reviewer 3 Report
The manuscript “Primitive compartmentalization for the sustainable replication 2 of genetic molecules” reviews several types of compartments to repress the amplification of parasitic molecules in prebiotic evolution. Both the theoretical and experimental studies are covered. Interesting and relevant topic in this field is presented and discussed.
There remain several points that might be clarified:
1. In lines 4-5. Are there any strategies except compartmentalization to avoid the replication of parasite RNA? The appropriate citations might be provided.
2. In lines 61. RNA. The biochemical characters of functional RNA used for these experience and simulation are unclear. Nucleotide sequence, size, GC ratio, 2ndary structure, mutation rate, affinity to the enzymes for replication, concentration in the liquid and others. Brief explanations will help readers.
3. Table1. The parasite RNA diffusion is limited. On the other hand, substrate (NTP) diffusion is free. Is this a realistic assumption? How does the diffusion are controlled?
Author Response
Please see the attached pdf file.
